# The evolution of uncertainty of learning in games

**Yun Kuen Cheung**[*]
Royal Holloway
University of London

**Georgios Piliouras**[*]
Singapore University of
Technology and Design

**Yixin Tao**[*]
London School of
Economics

## ABSTRACT

Learning-in-games has become an object of intense interest for ML due to its connections to numerous AI architectures. We study standard online learning in games but from a non-standard perspective. Instead of studying the behavior of a single initial condition and whether it converges to equilibrium or not, we study the behavior of a *probability distribution/measure over a set of initial conditions*. This initial uncertainty is well motivated both from a standard game-theoretic perspective (e.g. a modeler's uncertainty about the agents' initial beliefs, random external signals) as well as from a ML one (e.g. noisy measurements, system initialization from a dataset distribution). Despite this, little is formally known about whether and under what conditions uncertainty is amplified or reduced in these systems. We use the popular measure of differential entropy to quantify the evolution of uncertainty. We find that such analysis shares an intimate relationship with volume analysis, a technique which was recently used to demonstrate the occurrence of Lyapunov chaos when using Multiplicative Weights Update (MWU) or Follow-the-Regularized-Leader (FTRL) algorithms in zero-sum games. This allows us to show that the differential entropy of these learning-in-game systems increases linearly with time, formalizing their increased unpredictability over time. We showcase the power of the framework by applying it in the study of multiple related systems, including different standard online optimization algorithms in numerous games and dynamics of evolutionary game theory.

## 1  INTRODUCTION

A primary goal of ML research is to understand the behavior of learning algorithms in various settings. One standard approach is to determine from each initial condition whether a learning algorithm converges to a local optimum or stable state. Yet, in the context of online learning in games, and more generally in distributed multi-agent learning, it is natural to consider the evolution of a probability distribution over initial conditions instead. In these settings, each agent forms an initial belief on her own, and she typically does not reveal her belief to other agents or external modelers/analysts. For a modeler to understand the possible behaviors of the system while being uncertain of the agents' beliefs, one natural approach is to infer how the initialization distributes by using data of observations from the past. The modeler then uses this inference to predict the likelihoods of different outcomes in the future, either by simulation or by analysis. Also, random initialization can happen due to random external signals, e.g. weather, as well as noisy measurements. For readers who wish to learn more mathematical aspects and intuition behind of such models, see Appendix A.

In such cases, it is critical to understand whether the initial probability distribution evolves toward stability, or if its uncertainty gets amplified. Such analysis provides insight into the learning system's robustness against random initialization and environmental perturbations. If a system coordinator desires stability but the analysis shows the uncertainty gets amplified, she ought to coordinate with the agents to make changes, e.g. encourage them to use other online learning algorithms.

To analyze how uncertainty evolves, we need a measure of it. A natural choice is *entropy*. In his seminal work in 1948, Claude Shannon formulated an axiomatic foundation of information theory and introduced the seminal definition of Shannon entropy (SE): given a *discrete* random variable with possible outcomes $x_1, \ldots, x_n$ which occur with probability $p_1, \ldots, p_n$, its SE is

---

[*]E-mails: `yunkuen.cheung@rhul.ac.uk`, `georgios@sutd.edu.sg`, `Y.Tao16@lse.ac.uk`

$-\sum_{i=1}^{n} p_i \log p_i = \mathbb{E}\left[\log(1/p_i)\right]$. Entropy is the canonical measure of uncertainty: when $p_i = 1$ for some $i$, the distribution is considered certain and its entropy is zero; uniform distribution is viewed as the most uncertain, and it attains the maximum possible entropy of $\log n$. For *continuous* random variables with probability density function $g$, Shannon proposed an extension of SE called the *differential entropy* (DE), defined as $\mathbb{E}\left[\log(1/g(x))\right] = -\int_{\mathcal{X}} g(x) \log g(x)\,\mathrm{d}x$, where $\mathcal{X}$ is the support set of the random variable. *We will analyze how DE evolves in multi-agent learning.*

**Our Contributions.** In our model, a learning-in-game system starts with a distribution over a set of initial conditions in the *cumulative payoff space*, a coordinate system which is inherent in the implementations of many online learning algorithms. The initial distribution evolves over time depending on the combination of agents' learning algorithms and the underlying game. We focus on two popular learning algorithms, Multiplicative Weights Update (MWU) and its optimistic variant (OMWU).[1] The game settings include the standard two-player matrix games, and one-population games which are fundamental in biological evolution models. We show that the DE of a broad range of learning-in-game systems increases linearly with time (up to a certain limit), formalizing their increased unpredictability. Such systems include MWU in zero-sum games, OMWU in coordination games, and MWU in certain one-population games which reward more to intra-species play than to inter-species play. Our results apply to any initial distribution of absolutely continuous random vectors with finite differential entropy.

At this point one may naturally wonder: What level of uncertainty does a linear increase of DE indicate? What are its implications? To answer these questions, it is best to compare ourselves against other simple benchmarks. Consider the following simple learning system: in each round, the payoffs for each action are generated from a uniform distribution over $[-1, 1]$. In the cumulative payoff space, the distribution at time $T$ converges to a multivariate Gaussian distribution with variance $\Theta(T)$, and hence the entropy grows at a rate of $\mathcal{O}(\log T)$, much slower than linear growth. In Section 4.1, we present an implication of linear DE growth in learning-in-game systems. Briefly speaking, the DE cannot increase substantially for an indefinite amount of time in such systems. Thus, the distribution after a sufficiently long time must concentrate in a region that yields slower DE growth or decline; such region does not cover any point in the cumulative payoff space that corresponds to an interior Nash equilibrium. We will refer to this phenomenon by "the grand escape". See Theorem 4.5 for the formal description of the grand escape phenomenon; an additional assumption, namely the initial distribution has bounded support, is needed to establish this theorem. For the implications in information theory, we refer readers to Chapter 8 of Cover & Thomas (2006).

The central tool in analyzing the changes of DE is the *Jacobian matrix* of our multi-agent dynamical system. This is also the key notion in *volume analysis*, which was used in a recent series of papers to demonstrate Lyapunov chaos in learning-in-game systems. (Cheung & Piliouras (2019; 2020); Cheung & Tao (2020)) By showing that the determinant of the Jacobian matrix is strictly above 1 in a large domain of the cumulative payoff space, they showed that the volume (Lebesgue measure) of a small set of initial conditions will blow up exponentially to become a set of large diameter, thus demonstrating a general Lyapunov chaos phenomenon in learning-in-game. Indeed, the same property of Jacobian matrix guarantees linear increase of DE. Additionally, we present the first Jacobian analysis of online learning in one-population games. Finally, our DE analysis is robust against small perturbations. Consequently, our results extend to the setting where the game in each round can be different, as long as the payoffs are perturbations of each other. Such settings capture many games in the real world, where we know the "reference values" of the payoffs, but the accurate payoffs in each round differ slightly due to unknown external (random or adversarial) factors.

Our model and analysis can be viewed as an a strengthening of volume analysis. Volume analysis focuses on whether it is *possible* to reach a state or not, whereas in our model we are also concerned with how *probable/likely* it is to reach a state. More explicitly, a state can be reached but only with a tiny chance. When studying chaos and volume, such states matter, but less so when studying uncertainty as their contributions to entropy will be low. To illuminate this, we simulate MWU (with step-size 0.01) in Matching Pennies game, and present a few plots in Figure 2. The top-left plot shows the initial set, which is a small square around the unique Nash equilibrium. The top-right plot

---

[1]As we shall point out in Section 3, all our results about MWU also extend easily to the broad family of Follow-the-Regularized-Leader (FTRL) algorithms, which is a generalization of MWU. For clarity of our presentation, we choose to focus on MWU in this paper.

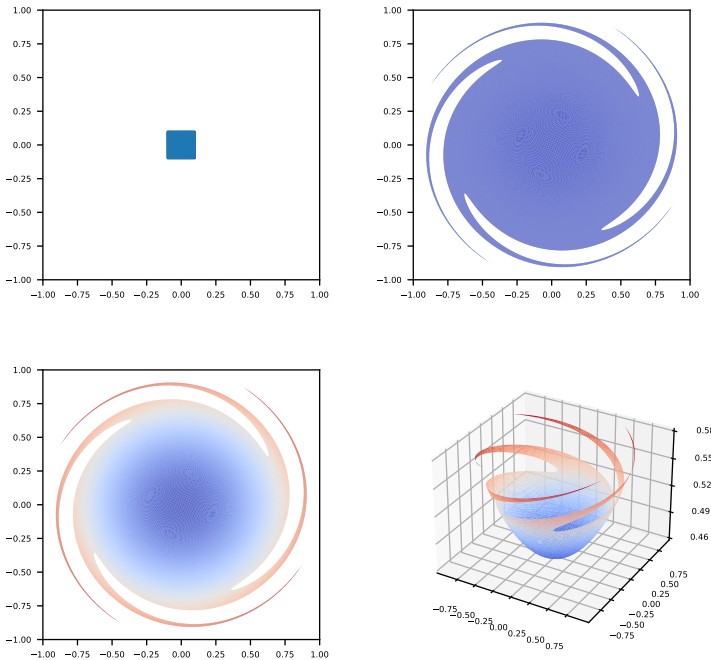

Figure 1: "Possible" vs. "Probable": In the heat-map, red (warm) and blue (cool) colors represent high and low densities respectively.

shows the range of possibility after 40000 steps[2]. In our model, we assume the initial distribution is uniform over the small square, and plot the heat-map of probability densities after 40000 steps (bottom-left). We observe that the states in the boundary of the vortex are more probable to occur, while the densities around the Nash equilibrium decline. The bottom-right plot shows the densities that generate the heat-map.

**Further Related Work.** Learning models that explicitly study the effects random initialization have received relatively little attention in game theory where static equilibrium concepts are typically the preferred solution. Lahkar & Seymour (2013) studies online learning under random initialization in population dynamics, where one population game is a common payoff model. In population dynamics, there exists a large population of agents who have different initial choices of mixed strategies, modeled as a distribution. The dynamics proceed by randomly pairing up agents in each round to play a game. Lahkar & Seymour (2014) extends this framework to other learning dynamics to describe the evolution of a distribution of mixed strategies in a population playing variants of Rock-Paper-Scissors games establishing local convergence to the interior equilibrium. Recently, further learning models inspired by Q-learning dynamics have been defined (Hu et al., 2019). Very little is formally known about the behavior of such models particularly under instability.

For zero-sum games with an interior Nash equilibrium, "the grand escape" phenomenon follows from Bailey & Piliouras (2018); Cheung (2018), who showed that MWU diverges away from Nash. Our results are more general, since their analysis only works with those zero-sum game dynamics, while our technique applies to many other game dynamics as well. Such instability results in zero-sum games are undesirable for ML applications such as GANs thus a lot of effort has been invested in developing algorithms that achieve convergence Daskalakis et al. (2017); Mertikopoulos et al. (2019); Daskalakis & Panageas (2018); Gidel et al. (2019); Mokhtari et al. (2020); Perolat et al. (2021). Our research direction here is in some sense orthogonal. We aim to characterize in a more detailed fashion the behavior of learning dynamics despite their instability.

The instability of MWU, FTRL and other optimization-driven dynamics in games has attracted a lot of attention in recent years. One formal technique to establish the unpredictability of MWU

---

[2]This phenomenon was first discussed in Cheung & Piliouras (2019) and called the "von Neumann vortex".

and other dynamics is based on proving Li-Yorke chaos, which has been established robustly in different families of routing/potential/Cournot games (Palaiopanos et al. (2017); Chotibut et al. (2020); Bielawski et al. (2021); Cheung et al. (2021)). The techniques in those papers are independent from ours as they require the identification of one dimensional invariant manifolds, and Li-Yorke chaos implies the existence of periodic orbits, which is not possible in our systems. In a very recent series of works, Flokas et al. (2020); Giannou et al. (2021) established that *all* (even partially) mixed equilibria in *all* games are *not* asymptotically stable for *any choice* for FTRL dynamics both in discrete as well as in the idealized continuous-time limit. In fact, it is possible to construct simple matrix games such that the orbits of the resulting dynamics can be arbitrarily complex Andrade et al. (2021). If, inspired by ML applications, we allow for more complex differentiable games it becomes even simpler to establish strong instability results for effectively any type of training algorithms (Letcher (2020); Balduzzi et al. (2020)). The sweeping nature of these strong negative results showcases the practical interest in uncovering more information about the nature of unstable dynamics of games. Our proposed framework introduces a novel, quantitative methodology for their study.

The notion of differential entropy (DE) has been used as uncertainty measure in many works across multiple disciplines. We note that while the extended definition of DE seems natural, it is not considered as the perfect generalization of SE since it misses some favorable properties, and therefore, as quoted from the popular information theory text of Cover & Thomas (2006), "there is need for some care in using the concept".[3] Nevertheless, DE remains an effective measure of uncertainty[4], which is commonly used in physics, economics, control theory, engineering, biology, medical research and beyond. For a general discussion, we refer readers to the text of Cover & Thomas (2006). For background information of evolutionary game theory, we recommend Hofbauer & Sigmund (1998); Sandholm (2010); Vega-Redondo (1996).

## 2 MODEL

We use bold small letters to denote vectors, and bold capital letters to denote matrices. Let $\Delta^d$ denote the probability simplex of dimension $d$: $\Delta^d := \{\mathbf{x} \in \mathbb{R}^d : \text{for each } j, \ x_j \geq 0, \text{ and } \sum_{j=1}^d x_j = 1\}$.

**MWU and OMWU.** When an agent uses MWU, she wants to choose among $d \geq 2$ *pure actions* based on the past cumulative payoffs to these actions. The process starts at discrete time $t = 0$, where the agent chooses an *initial cumulative payoff vector* $\mathbf{p}^0 \in \mathbb{R}^d$. At any time $t \geq 1$, upon receiving the payoffs to the each of the $d$ actions at time $t - 1$, which are denoted by a vector $\mathbf{r}^{t-1} \in \mathbb{R}^d$, the agent updates the cumulative payoff vector by the rule

$$\forall j = 1, 2, \ldots, d, \quad p_j^t \leftarrow p_j^{t-1} + \epsilon \cdot r_j^{t-1}, \tag{1}$$

where $\epsilon > 0$ is the step-size of the update. At any time $t \geq 0$, the agent chooses randomly among the $d$ pure actions depending on the value of $\mathbf{p}^t$. Precisely, the agent chooses a *mixed strategy* $\mathbf{x}^t \in \Delta^d$ which is a function of $\mathbf{p}^t$. For MWU, $\mathbf{x}^t$ is determined by the rule

$$\forall j = 1, 2, \ldots, d, \quad x_j^t \leftarrow \exp(p_j^t) / \left( \sum_{k=1}^d \exp(p_k^t) \right). \tag{2}$$

For OMWU, we start with $\mathbf{p}^0 = \mathbf{p}^1$, and for $t \geq 2$,
$$\forall j = 1, 2, \ldots, d, \quad p_j^t \leftarrow p_j^{t-1} + \epsilon \cdot (2r_j^{t-1} - r_j^{t-2}), \tag{3}$$

and $\mathbf{x}^t$ is determined by the same rule (2).

In general, the payoff vectors $\mathbf{r}^{t-1}$ can be arbitrary, and they may or may not depend on the mixed strategy $\mathbf{x}^{t-1}$. However, in the context of learning-in-game, the underlying game generates the payoff vectors that depend on the mixed strategies of the agents involved. We discuss two popular game models relevant to our work, namely *two-player matrix games* and *one-population games*.

---

[3]Two issues with DE are: (1) DE can be negative and indeed it has no finite lower bound, while SE is always positive; (2) DE depends on the choice of coordinate system. The positivity of SE is favorable in information theory where they want to measure information contents of different communications. But for the purpose of measuring and comparing uncertainty, positivity does not seem relevant. (2) is not an issue for us, since we will always stick to a fixed coordinate system which is natural in learning-in-games.

[4]DE captures uncertainty well, e.g. for all popular distributions (Gaussian, uniform, exponential, Rayleigh etc), their DE increases with their variances. Also, among all distributions over a bounded support set, the uniform distribution over the support set attains maximum DE.

**Two-Player Matrix Games.** In a two-player matrix game, Player 1 has $n$ actions and Player 2 has $m$ actions. The game is specified by two matrices $\mathbf{A}, \mathbf{B} \in [-1,1]^{n \times m}$. When Player 1 chooses action $j$ and Player 2 chooses action $k$, $A_{jk}, B_{jk}$ are the payoffs to Players 1 and 2 respectively. When the players choose mixed strategies, the payoffs are extended via expectation. Denote the mixed strategies of Players 1 and 2 by $\mathbf{x} \in \Delta^n$ and $\mathbf{y} \in \Delta^m$. The payoff to action $j$ of Player 1 is $[\mathbf{A}\mathbf{y}]_j$, while the payoff to action $k$ of Player 2 is $[\mathbf{B}^\mathsf{T}\mathbf{x}]_k$. When both players use MWU or OMWU to play this game repeatedly, we have the following discrete-time dynamical system, where $\mathbf{p}^t, \mathbf{q}^t$ denote the cumulative payoff vectors at time $t$ of Players 1 and 2 respectively. The two players choose initial cumulative payoff vectors $\mathbf{p}^0, \mathbf{q}^0$.

$$\forall t \geq 1, \quad \mathbf{p}^t \ \leftarrow \ \mathbf{p}^{t-1} + \epsilon \cdot \mathbf{A}\mathbf{y}^{t-1};$$
$$\mathbf{q}^t \ \leftarrow \ \mathbf{q}^{t-1} + \epsilon \cdot \mathbf{B}^\mathsf{T}\mathbf{x}^{t-1}. \tag{4}$$

Note that $\mathbf{x}^{t-1}$ is a function of $\mathbf{p}^{t-1}$ as specified in (2); similarly, $\mathbf{y}^{t-1}$ is a function of $\mathbf{q}^{t-1}$. Thus, we can also view (4) as an iterated function that maps from $(\mathbf{p}^{t-1}, \mathbf{q}^{t-1})$ to $(\mathbf{p}^t, \mathbf{q}^t)$.

In the above setting, the game $(\mathbf{A}, \mathbf{B})$ is the same at all times $t$. It is also interesting to consider scenarios where the game varies in each round. Our results extend to settings where there is a "reference game" $(\mathbf{A}, \mathbf{B})$, but at each round the actual game being played is a perturbation of the reference game. Precisely, the game at time $t$ is $(\mathbf{A}^t, \mathbf{B}^t) = (\mathbf{A}, \mathbf{B}) + (\boldsymbol{\Delta}_A^t, \boldsymbol{\Delta}_B^t)$, where $\boldsymbol{\Delta}_A^t, \boldsymbol{\Delta}_B^t$ are matrices with maximum absolute entry at most $\beta$, for some small $\beta > 0$.

**One-Population Games.** One-population games is a fundamental model in mathematical biology, which is widely used to explain the evolution of a population of different of species over time. An one-population game is similar to a two-player game, but now Player 1 is playing against herself; there is only one player (or in biology term, one population) in this game, while a mixed strategy $\mathbf{x}^t$ represents the fraction of different species among the population. An one-population game is specified by a matrix $\mathbf{A} \in \mathbb{R}^{n \times n}$. When the mixed strategy of the player is $\mathbf{x} \in \Delta^n$, the payoff to action $j$ is $[\mathbf{A}\mathbf{x}]_j$. The resultant discrete-time dynamical system with MWU is

$$\forall t \geq 1, \quad \mathbf{p}^t \ \leftarrow \ \mathbf{p}^{t-1} + \epsilon \cdot \mathbf{A}\mathbf{x}^{t-1}, \tag{5}$$

where $\mathbf{x}^{t-1}$ is a function of $\mathbf{p}^{t-1}$ as in (2).

**Random Initial Cumulative Payoff Vectors.** In Section 1, we explained why we consider uncertainty of the initial cumulative payoff vectors. This can be captured by a model where these vectors follow a distribution. We use two-player game for discussion below; for one-population game it is similar. Assume that the initial vectors $(\mathbf{p}^0, \mathbf{q}^0)$ follow a joint probability distribution in the domain $\mathbb{R}^n \times \mathbb{R}^m$. As the initial vectors are random and they are updated according to the discrete-time dynamical system (4), for any $t \geq 0$, $(\mathbf{p}^t, \mathbf{q}^t)$ also follows a distribution, which is the main object we analyze in this work. We are interested in knowing certain statistics of uncertainty of this distribution, say differential entropy, which is the subject of the next section.

## 3 Differential Entropy

In this section, we discuss how changes of differential entropy (DE) can be analyzed in general, and provide some intuition of the relationship between DE and volume. To proceed, we need a crucial notion called the *Jacobian matrix*. Let $f : \mathbb{R}^d \to \mathbb{R}^d$ be a smooth function. Its Jacobian matrix is

$$\mathbf{J}_f = \begin{bmatrix} \frac{\partial f_1}{\partial x_1} & \frac{\partial f_1}{\partial x_2} & \cdots & \frac{\partial f_1}{\partial x_d} \\ \vdots & \vdots & \ddots & \vdots \\ \frac{\partial f_d}{\partial x_1} & \frac{\partial f_d}{\partial x_2} & \cdots & \frac{\partial f_d}{\partial x_d} \end{bmatrix}.$$

We also need multivariate integration by substitution; see Appendix B for a discussion of it.

Let $X$ be an absolutely continuous random variable in $\mathbb{R}^d$. Let $g$ be its probability density function, whose support set is $\mathcal{X}$. The DE of $X$ is $h(X) := -\int_{\mathcal{X}} g(x) \log g(x) \, \mathrm{d}x$, which we assume to be finite. For any function $f : \mathbb{R}^d \to \mathbb{R}^d$, $f(X)$ is another random variable; we denote its probability density function by $\hat{g}$, and its support set by $\mathcal{Y}$. If $f$ is a diffeomorphism, using integration

by substitution, we have $\hat{g}(y) = g(f^{-1}(y)) \cdot \left|\det \mathbf{J}_{f^{-1}}(y)\right|$. Using integration by substitution again, the DE of $f(X)$ can be computed as follows:

$$
\begin{aligned}
h(f(X)) &= -\int_{\mathcal{Y}} \hat{g}(y) \log \hat{g}(y) \mathsf{d}y \\
&= -\int_{\mathcal{Y}} g(f^{-1}(y)) \cdot \left|\det \mathbf{J}_{f^{-1}}(y)\right| \cdot \log\left(g(f^{-1}(y)) \cdot \left|\det \mathbf{J}_{f^{-1}}(y)\right|\right) \mathsf{d}y \\
&= -\int_{\mathcal{X}} g(x) \cdot \left|\det \mathbf{J}_{f^{-1}}(f(x))\right| \cdot \log\left(g(x) \cdot \left|\det \mathbf{J}_{f^{-1}}(f(x))\right|\right) \cdot \left|\det \mathbf{J}_f(x)\right| \mathsf{d}x. \quad (6)
\end{aligned}
$$

By the Inverse Function Theorem, $\left|\det \mathbf{J}_{f^{-1}}(f(x))\right| \cdot \left|\det \mathbf{J}_f(x)\right| = 1$. Thus,

$$
h(f(X)) - h(X) = \int_{\mathcal{X}} g(x) \log \left|\det \mathbf{J}_f(x)\right| \mathsf{d}x \quad (7)
$$

when the integral on the RHS is finite. This is true if $\left|\det \mathbf{J}_f(x)\right|$ is a differentiable and bounded function of $x$, which holds in all learning-in-game systems we study. Thus, the Jacobian's determinant directly affects the change of DE. If there exists $\alpha > 0$ such that $\left|\det \mathbf{J}_f(x)\right| \geq 1 + \alpha$ for all $x \in \mathcal{X}$, then $h(f(X)) - h(X) \geq \log(1 + \alpha) > 0$. In order to use (7) to analyze the changes of DE of learning in two-player matrix games, we view the dynamical system (4) as a function that maps from $(\mathbf{p}^{t-1}, \mathbf{q}^{t-1})$ to $(\mathbf{p}^t, \mathbf{q}^t)$ in each round $t \geq 1$. If the change of DE per round is at least $\log(1 + \alpha) > 0$, then the DE increases linearly with time.

**The Relationship Between DE and Volume.** Before discussing the relationship, we first recap what is volume. Let $\mathcal{X} \subset \mathbb{R}^d$ be a measurable set with positive Lebesgue measure (which is sometimes called the volume of the set), and let $f : \mathbb{R}^d \to \mathbb{R}^d$ be a diffeomorphism. The volume of $f(\mathcal{X})$ is $\int_{\mathcal{X}} \left|\det \mathbf{J}_f(x)\right| \mathsf{d}x$. When $\left|\det \mathbf{J}_f(x)\right| \geq 1 + \alpha$ for all $x \in \mathcal{X}$, the volume increases by a factor of at least $(1+\alpha)$. If this occurs in a dynamical system every round, the volume of the initial set grows exponentially with time, and hence also its diameter, establishing that Lyapunov chaos occurs.

The occurrence of the Jacobian in both the formulae for computing volume and the change of DE in (7) is not a coincidence, but with a simple intuition behind. The Jacobian yields a linear approximation of $f$ locally: for any $x \in \mathbb{R}^d$ and any small perturbation vector $\Delta x \in \mathbb{R}^d$, $f(x + \Delta x) \approx f(x) + \mathbf{J}_f(x) \cdot \Delta x$. We consider a tiny hypercube with volume $v$ around $x$, and its image under the function $f$. By the linear approximation, the image is approximately a parallelotope (the high-dimensional analogue of parallelogram), whose volume is known to be $\left|\det \mathbf{J}_f(x)\right| \cdot v$. By partitioning $\mathcal{X}$ into infinitesimal tiny hypercubes and summing up, this is intuitively how the integral formula for computing volume is obtained. Next, suppose the probability density at $x$ is $g(x)$. The probability of the tiny hypercube is approximately $g(x) \cdot v$. After the transformation $f$, this amount of probability is spread approximately uniformly in the parallelotope, thus the new density at $f(x)$ is $g(x)/\left|\det \mathbf{J}_f(x)\right|$. If the volume increases locally at $x$, i.e. $\left|\det \mathbf{J}_f(x)\right| > 1$, it intuitively means the uncertainty level increases locally. This intuition is captured by DE: the contribution to DE by the tiny hypercube is $-g(x) \cdot v \log(g(x) \cdot v)$, while the contribution to DE by the parallelotope is $-g(x) \cdot v \log \frac{g(x) \cdot v}{\left|\det \mathbf{J}_f(x)\right|}$. Thus, the local change of DE is $g(x) \cdot v \log \left|\det \mathbf{J}_f(x)\right|$, which has the same sign as $(\left|\det \mathbf{J}_f(x)\right| - 1)$. To summarize, *local volume increase is equivalent local DE increase*.

This equivalence allows us to spot that for those learning systems in two-player matrix games studied in the series of volume analysis papers of Cheung & Piliouras (2019; 2020); Cheung & Tao (2020), their DE with our model increases linearly in a large domain of the cumulative payoff space. We present this result formally Section 4.1, and present an interesting consequence of the DE growth. Partly motivated by Lahkar & Seymour (2013), we also transfer the techniques to analyzing one-population games in Section 4.2, and we present two sufficient conditions on these games that leads to linear DE growth with MWU.

As first spotted by Cheung & Piliouras (2019), these results not only cover MWU and OMWU, but also the broad family of FTRL algorithms, since the properties of the Jacobian for FTRL is very similar to that of MWU. (This is not surprising since FTRL is a generalization of MWU.)

## 4 APPLICATIONS

In this section, we consider two applications: MWU in two-player zero-sum games and one-population games. In Appendix C, we also consider OMWU dynamics in two-player coordination

games and one-population games, and the settings where the game in each round is perturbed. A two-player game $(\mathbf{A}, \mathbf{B})$ is *zero-sum* if $\mathbf{A} = -\mathbf{B}$. We focus on non-trivial games, defined below.

**Definition 4.1 (Cheung & Piliouras (2019))** *A zero-sum game $(\mathbf{A}, -\mathbf{A})$ is trivial if there exists real numbers $a_1, a_2, \cdots, a_n$ and $b_1, b_2, \cdots, b_m$ such that $A_{jk} = a_j + b_k$. The same definition applies for one-population game $\mathbf{A}$.*

Cheung & Piliouras pointed out that a trivial game is not interesting as the players have dominant strategies. They provided a measure of distance of a zero-sum game $(\mathbf{A}, -\mathbf{A})$ from triviality by

$$c(\mathbf{A}) = \min_{a_1, \cdots, a_n, b_1, \cdots, b_m} [\max(A_{jk} - a_j - b_k) - \min(A_{jk} - a_j - b_k)] \qquad (8)$$

### 4.1 MWU IN TWO-PLAYER ZERO-SUM GAME

Recall from (4) that in zero-sum game $(\mathbf{A}, -\mathbf{A})$, MWU dynamic in the cumulative payoff space is

$$\mathbf{p}^{t+1} = \mathbf{p}^t + \epsilon \cdot \mathbf{A}\mathbf{y}(\mathbf{q}^t) \quad \text{and} \quad \mathbf{q}^{t+1} = \mathbf{q}^t + \epsilon \cdot (-\mathbf{A})^\top \mathbf{x}(\mathbf{p}^t)$$

where $\mathbf{x}(\mathbf{p}^t)$ and $\mathbf{y}(\mathbf{q}^t)$ are the mixed strategies of two players at time $t$: $x_j(\mathbf{p}^t) = \exp(p_j^t)/\sum_{j'} \exp(p_{j'}^t)$ and $y_k(\mathbf{q}^t) = \exp(q_k^t)/\sum_{k'} \exp(q_{k'}^t)$. Two different regions are considered: the *interior region* $S_1 := \{(\mathbf{p}, \mathbf{q}) \mid \mathbf{x}(\mathbf{p}) \geq \delta \text{ and } \mathbf{y}(\mathbf{q}) \geq \delta\}$, and the *boundary region* $S_2$, which is the complement to $S_1$. Note that if the game possesses an interior Nash equilibrium, then the equilibrium lies in $S_1$ for any sufficiently small $\delta$.

Let $g^t(\mathbf{p}, \mathbf{q})$ denote the probability density of $(\mathbf{p}, \mathbf{q})$ at round $t$. Recall from 7 that the change of entropy is $\int g^t(\mathbf{p}, \mathbf{q}) \log |\det \mathbf{J}_f(\mathbf{p}, \mathbf{q})| \, \mathsf{d}(\mathbf{p}, \mathbf{q})$. If $g^t(\mathbf{p}, \mathbf{q})$ is fully supported by $S_1$, then by using the Jacobian analysis of Cheung & Piliouras (2019), who showed that the determinant of the Jacobian matrix is strictly larger than 1 in $S_1$, we show the differential entropy increases linearly.

**Theorem 4.2** *For any non-trivial two-player zero-sum game $(\mathbf{A}, -\mathbf{A})$, with MWU dynamics of step-size $\epsilon \leq \min\{1/(32n^2m^2), \delta^2 c(\mathbf{A})^2/8\}$, if $g^t(\mathbf{p}, \mathbf{q}) > 0$ only for $(\mathbf{p}, \mathbf{q}) \in S_1$, then the differential entropy will increase by at least $\frac{\delta^2 c(\mathbf{A})^2 \cdot \epsilon^2/8}{1 + \delta^2 c(\mathbf{A})^2 \cdot \epsilon^2/8}$ in round $t$.*

**Proof:** For MWU dynamics, $f(\mathbf{p}, \mathbf{q}) = (\mathbf{p} + \epsilon \mathbf{A}\mathbf{y}(\mathbf{q}), \mathbf{q} - \epsilon \mathbf{A}^\top \mathbf{x}(\mathbf{p}))$ by (2). Cheung & Piliouras (2019) showed that if $\epsilon \leq \min\{1/(32n^2m^2), \delta^2 c(\mathbf{A})^2/8\}$, then $\det \mathbf{J}_f(\mathbf{p}, \mathbf{q}) \geq 1 + \frac{\delta^2 c(\mathbf{A})^2}{8} \cdot \epsilon^2$ for any $(\mathbf{p}, \mathbf{q}) \in S_1$, where $c(\mathbf{A})$ is the measure of the distance of a zero-sum game $(\mathbf{A}, -\mathbf{A})$ from triviality (8). As $\log(1+r) \geq \frac{r}{1+r}$ for any $r > -1$, $\int g^t(\mathbf{p}, \mathbf{q}) \log |\det \mathbf{J}_f(\mathbf{p}, \mathbf{q})| \, \mathsf{d}(\mathbf{p}, \mathbf{q}) \geq \frac{\delta^2 c(\mathbf{A})^2 \cdot \epsilon^2/8}{1 + \delta^2 c(\mathbf{A})^2 \cdot \epsilon^2/8}$. $\qquad \square$

Actually, the assumption that $g^t(\mathbf{p}, \mathbf{q})$ is fully supported in $S_1$ can be weakened. We show that if a sufficiently large probability lies in $S_1$, then the differential entropy still increases linearly. With this result, we can show Theorem 4.5 below, which states that there exists infinitely many rounds, such that in each of these rounds $t$, the probability of $(\mathbf{p}^t, \mathbf{q}^t) \in S_2$ will be at least some constant.

**Theorem 4.3** *If $\int_{S_1} g(\mathbf{p}, \mathbf{q}) \, \mathsf{d}(\mathbf{p}, \mathbf{q}) \geq \lambda$, then the entropy will increase by at least $\frac{\delta^2 c(\mathbf{A})^2 \cdot \epsilon^2/8}{1 + \delta^2 c(\mathbf{A})^2 \cdot \epsilon^2/8} \cdot \lambda - \frac{\epsilon^3}{1-\epsilon^3} \cdot (1 - \lambda)$ in round $t$, which is strictly positive whenever $\lambda > \frac{16\epsilon}{\delta^2 c(\mathbf{A})^2}$.*

**Proof:** Cheung & Piliouras (2019) showed that if $\epsilon \leq \min\{1/(32n^2m^2), \delta^2 c(\mathbf{A})^2/8\}$, then $\det \mathbf{J}_f(\mathbf{p}, \mathbf{q}) \geq 1 + \frac{\delta^2 c(\mathbf{A})^2}{8} \cdot \epsilon^2$ for any $(\mathbf{p}, \mathbf{q}) \in S_1$, and $\det \mathbf{J}_f(\mathbf{p}, \mathbf{q}) \geq 1 - \epsilon^3$ for any $(\mathbf{p}, \mathbf{q}) \in S_2$. Recall that the change of entropy is $\int g^t(\mathbf{p}, \mathbf{q}) \log |\det \mathbf{J}_f(\mathbf{p}, \mathbf{q})| \, \mathsf{d}(\mathbf{p}, \mathbf{q})$. As $\log(1+r) \geq \frac{r}{1+r}$ for any $r > -1$, the change of entropy is at least

$$\int_{S_1} g^t(\mathbf{p}, \mathbf{q}) \log \left(1 + \frac{\delta^2 c(\mathbf{A})^2}{8} \cdot \epsilon^2\right) \mathsf{d}(\mathbf{p}, \mathbf{q}) + \int_{S_2} g^t(\mathbf{p}, \mathbf{q}) \log(1 - \epsilon^3) \, \mathsf{d}(\mathbf{p}, \mathbf{q})$$

$$\geq \frac{\delta^2 c(\mathbf{A})^2 \cdot \epsilon^2/8}{1 + \delta^2 c(\mathbf{A})^2 \cdot \epsilon^2/8} \int_{S_1} g^t(\mathbf{p}, \mathbf{q}) \, \mathsf{d}(\mathbf{p}, \mathbf{q}) - \frac{\epsilon^3}{1 - \epsilon^3} \int_{S_2} g^t(\mathbf{p}, \mathbf{q}) \, \mathsf{d}(\mathbf{p}, \mathbf{q}). \qquad \square$$

In Theorem 4.3, for any fixed $\delta$ and $c(\mathbf{A})$, the lower bound of $\lambda$ decreases with $\epsilon$. In particular, if $\epsilon = \delta^2 c(\mathbf{A})^2/64$, the lower bound is $1/4$, i.e. it only requires a small probability in $S_1$ for the DE

to strictly increase. Theorem 4.3 indicates that the DE growth is at least linear ($\Omega(T)$) as long as the distribution has a sufficient amount of probability lying in $S_1$. Next, in Lemma 4.4 we show that the DE growth is at most logarithmic ($\mathcal{O}(\log T)$) whenever the initial distribution has bounded support. These two seemingly contradicting bounds can only be compatible for one reason: the condition needed by Theorem 4.3 will eventually be violated, i.e. there is insufficient amount of probability lying in $S_1$ after a sufficiently long time. We formalize this in Theorem 4.5 below.

**Lemma 4.4** *Suppose the initial distribution has bounded support in $[-b, b]^{n+m}$, for some $b > 0$. Then the differential entropy at time $T$ is at most $(n + m)\log(\epsilon T + b)$.*

**Proof:** Since every entry in $\mathbf{A}$ is bounded in the interval $[-1, 1]$, we have $|p_j^{t+1} - p_j^t| \leq \epsilon$ and $|q_k^{t+1} - q_k^t| \leq \epsilon$. Thus, the distribution at time $T$ has bounded support in $[-\epsilon T - b, \epsilon T + b]^{n+m}$. Within this box, the maximal differential entropy is attained by the uniform distribution over it, which is $(n + m)\log(\epsilon T + b)$. □

**Theorem 4.5** *For any non-trivial two-player zero-sum games $(\mathbf{A}, -\mathbf{A})$, with MWU dynamics of step-size $\epsilon \leq \min\{1/(32n^2m^2), \delta^2 c(\mathbf{A})^2/8\}$, for any initial distribution with bounded support, and for any $\lambda > \frac{16\epsilon}{\delta^2 c(\mathbf{A})^2}$, there exists an infinite sequence of times $(T_1, T_2, \cdots)$ such that the probability in $S_2$ at each round $T_i$ is at least $1 - \lambda$.*

The above theorem gives a lower bound of $1 - \lambda$ on the probability in the boundary area for infinitely many steps, which is the "grand escape" phenomenon we mentioned in the introduction.

## 4.2 MWU IN ONE POPULATION GAME

Recall from (5) that in one-population game $\mathbf{A}$, MWU dynamics in the cumulative payoff space is $\mathbf{p}^{t+1} = \mathbf{p}^t + \epsilon \cdot \mathbf{A}\mathbf{x}(\mathbf{p}^t)$, where $x_k(\mathbf{p}^t) = \frac{\exp(p_k^t)}{\sum_{k'} \exp(p_{k'}^t)}$.

Similar to the two-player zero-sum games, we also consider two different regions: the interior region $S_1 = \{\mathbf{p} \mid \mathbf{x}(\mathbf{p}) \geq \delta\}$, and the boundary region $S_2$, which is the complement to $S_1$. Let $g^t(\mathbf{p})$ denote the probability density of $\mathbf{p}$ at time $t$. We will show that for some sub-classes of one-population games $\mathbf{A}$, MWU dynamics increase differential entropy linearly under some conditions. Also we show the analogue of Theorem 4.5: there exists infinitely many rounds, such that at each of these rounds $t$, the probability of $\mathbf{p}^t \in S_2$ will be at least a constant. First of all, we present the Jacobian analysis of such systems.

**Jacobian Analysis.** The Jacobian of the dynamics is $\mathbf{J} = \mathbf{I} + \epsilon \cdot \mathbf{R}$, where $\mathbf{I}$ is the $n \times n$ identity matrix, and $\mathbf{R}$ is an $n \times n$ matrix, in which

$$R_{jk} = A_{jk}\frac{\exp(p_k)}{\sum_{k''}\exp(p_{k''})} - \sum_{k'}A_{jk'}\frac{\exp(p_{k'})\exp(p_k)}{\left(\sum_{k''}\exp(p_{k''})\right)^2} = x_k(\mathbf{p})\left(A_{jk} - \sum_{k'}A_{jk'}x_{k'}(\mathbf{p})\right).$$

For simplicity, with a little abuse of notation, we use $x_k$ as a shorthand of $x_k(\mathbf{p})$.

Next, we calculate the determinant of the Jacobian. When we compute it using the Leibniz formula, it can be expressed in the following form, which is a polynomial of $\epsilon$:

$$\det(\mathbf{J}(\mathbf{p})) = 1 + \epsilon \cdot (\text{first order term}) + \epsilon^2 \cdot (\text{second order term}) + \cdots,$$

where the first order term is

$$\sum_k R_{kk} = \sum_k x_k A_{kk} - \sum_{k,k'} x_k x_{k'} A_{kk'} = \frac{1}{2}\sum_{k \neq k'} x_k x_{k'}(A_{kk} + A_{k'k'} - A_{kk'} - A_{k'k}).$$

Thus, the following lemma gives the condition with which the first order term is non-negative. Note that in order to have a increasing differential entropy, non-negative first order term is a necessary condition for a small enough step size.

**Lemma 4.6** *The first order term is non-negative for all possible $\mathbf{x}$ if and only if $A_{kk} + A_{k'k'} \geq A_{kk'} + A_{k'k}$ for all $k, k'$. Moreover, if $A_{kk} + A_{k'k'} \geq A_{kk'} + A_{k'k}$ for all $k, k'$ and the inequality is strict for some $k, k'$, then the first order term is strictly positive for all $\mathbf{x} \geq \delta$.*

In the terminology of mathematical biology, $A_{kk}$ is the payoff to species $k$ for the intra-species play within the species, while $A_{kk'}$ is the payoff to species $k$ for the inter-species play between species $k$ and $k'$. Informally speaking, the condition in the above lemma is: the one-population game rewards intra-species play more than to inter-species play.

**Two Sufficient Conditions for Linear DE Growth.** Recall that to demonstrate linear DE growth, we want the determinant of the Jacobian to be strictly larger than 1. By Lemma 4.6, a clear sufficient condition is: $A_{kk} + A_{k'k'} \geq A_{kk'} + A_{k'k}$ for all $k$ and $k'$, and there exist $k, k'$ such that the inequality is strict. But when there is no $k, k'$ with strict inequality, it is still possible to have linear DE growth if the second order term is strictly positive. We present precise statements regarding these two cases below. To proceed, we define $s(\mathbf{A})$, which is positive for the first case: $s(\mathbf{A}) = \frac{1}{2} \cdot \max_{k,k'} \{A_{kk} + A_{k'k'} - A_{kk'} - A_{k'k}\}$.

For the first case, we now bound the higher order terms. Note that since $|A_{kk'}| \leq 1$ and hence $|R_{kk'}| \leq 2$. When expanding the determinant using the Leibniz formula, we get a polynomial of $\epsilon$. For any $i \geq 2$, each $\epsilon^i$ term in the expansion must be of the format $\epsilon^i$ times a product of $i$ factors of the form $R_{k_1,k_2}$, so each such term has absolute value bounded by $2^i \epsilon^i$. To count the number of such terms, note that within each term, the collection of $k_1$'s is same as the collection of $k_2$'s, but the order of $k_2$ can be any permutation of the $k_1$'s. A simple counting shows there are at most $\binom{n}{i} \cdot i! \leq n^i$ such terms, and hence the $i$-th order coefficient is bounded by $n^i \cdot 2^i$. Therefore, when $\epsilon \leq \frac{\delta^2 s(\mathbf{A})}{16n^2}$, $\det \mathbf{J}(\mathbf{p}) \geq 1 + \frac{\delta^2 s(\mathbf{A})}{2} \cdot \epsilon$ for all $\mathbf{p} \in S_1$, and $\det \mathbf{J}(\mathbf{p}) \geq 1 - 8n^2\epsilon^2$ for all $\mathbf{p} \in S_2$. With a similar argument as in two-player zero-sum game, we achieve the following theorems.

**Theorem 4.7** *If $\int_{S_1} g^t(\mathbf{p}) \, d\mathbf{p} \geq \lambda$, then the entropy will increase by at least $\frac{\delta^2 s(\mathbf{A}) \cdot \epsilon/2}{1 + \delta^2 s(\mathbf{A}) \cdot \epsilon/2} \cdot \lambda - \frac{8n^2\epsilon^2}{1 - 8n^2\epsilon^2} \cdot (1 - \lambda)$ in round $t$, which is strictly positive whenever $\lambda > \frac{32n^2\epsilon}{\delta^2 s(\mathbf{A})}$.*

**Theorem 4.8** *In a population game $\mathbf{A}$ such that $A_{kk} + A_{k'k'} \geq A_{kk'} + A_{k'k}$ for all $k$ and $k'$ and there exist $k$ and $k'$ such that the inequality is strict, with MWU dynamics of step-size $\epsilon \leq \frac{\delta^2 s(\mathbf{A})}{16n^2}$, for any initial distribution with bounded support, and for any $\lambda > \frac{32n^2\epsilon}{\delta^2 s(\mathbf{A})}$, there exists an infinite series of time $(T_1, T_2, \cdots)$ such that, for any $i$, the probability in $S_2$ at round $T_i$ is no less than $1 - \lambda$.*

In Appendix D, in the same spirit we did in Figure 2, we present a few plots of MWU in an one-population game that satisfies the condition in the above theorem.

Next, we consider the case that $A_{kk} + A_{k'k'} = A_{kk'} + A_{k'k}$ for all $k$ and $k'$. In this case, the first order term is 0. Therefore, we will focus on the second order term, which is $\sum_{k_1 < k_2}(R_{k_1 k_1} R_{k_2 k_2} - R_{k_1 k_2} R_{k_2 k_1})$, which upon expansion becomes

$$-\frac{1}{2}\left[\sum_{k_1,k_2} x_{k_1} x_{k_2} \left(A_{k_1 k_2} - \sum_{k'} A_{k_1 k'} x_{k'}\right)\left(A_{k_2 k_1} - \sum_{k'} A_{k_2, k'} x_{k'}\right)\right].$$

Let $C_{(\mathbf{A},\mathbf{B})}(\mathbf{x}, \mathbf{y}) = -\sum_{k_1,k_2} x_{k_1} y_{k_2}\left(A_{k_1 k_2} - \sum_{k'} A_{k_1 k'} y_{k'}\right)\left(B_{k_1 k_2} - \sum_{k'} B_{k',k_2} x_{k'}\right)$. Then the second order term is $\frac{1}{2} C_{(\mathbf{A},\mathbf{A}^\top)}(\mathbf{x}, \mathbf{x})$. We will show this term is non-negative.

**Lemma 4.9** *If $A_{kk} + A_{k'k'} = A_{kk'} + A_{k'k}$ for all $k$ and $k'$, then $\frac{1}{2} C_{(\mathbf{A},\mathbf{A}^\top)}(x,x) = \frac{1}{2} C_{(\mathbf{A},-\mathbf{A})}(x,x) \geq \frac{\delta^2 c(\mathbf{A})}{8}$.*

**Proof:** By Cheung & Tao (2020), $C_{(\mathbf{A},\mathbf{B})}(x,y) = C_{(\mathbf{A},\mathbf{B+T})}(x,y)$ if $T_{kk'} = u_k + v_{k'}$ for all $k$ and $k'$. As $(\mathbf{A}^\top - (-\mathbf{A}))_{ij} = A_{ji} + A_{ij} = A_{ii} + A_{jj}$, the first equality holds. The inequality follows from Cheung & Piliouras (2019). $\square$

Thus, for the determinant of the Jacobian, the first order term is 0 and the second order term is at least $\frac{\delta^2 c(\mathbf{A})}{8}$ in $S_1$. Moreover, the $i$-th order term is bounded by $n^i 2^i \epsilon^i$. Therefore, when $\epsilon \leq \frac{\delta^2 c(\mathbf{A})}{256n^3}$, $\det \mathbf{J}(\mathbf{p}) \geq 1 + \frac{\delta^2 c(\mathbf{A})}{16} \cdot \epsilon^2$ for any $\mathbf{p} \in S_1$, and $\det \mathbf{J}(\mathbf{p}) \geq 1 - 16n^3\epsilon^3$ for $x \in S_2$.

**Theorem 4.10** *If $\int_{S_1} g^t(\mathbf{p}) \, d\mathbf{p} \geq \lambda$ then the entropy will increase by at least $\frac{\delta^2 c(\mathbf{A}) \cdot \epsilon^2/16}{1 + \delta^2 c(\mathbf{A}) \cdot \epsilon^2/16} \cdot \lambda - \frac{16n^3\epsilon^3}{1 - 16n^3\epsilon^3} \cdot (1 - \lambda)$ in round $t$, which is strictly positive whenever $\lambda > \frac{512n^3\epsilon}{\delta^2 c(\mathbf{A})}$.*

**Theorem 4.11** *In a population game $\mathbf{A}$ such that $A_{kk} + A_{k'k'} = A_{kk'} + A_{k'k}$ for all $k$ and $k'$ and $\mathbf{A}$ is non-trivial, with MWU dynamics of step-size $\epsilon \leq \frac{\delta^2 c(\mathbf{A})}{256n^3}$, for any initial distribution with bounded support, and for any $\lambda > \frac{512n^3\epsilon}{\delta^2 c(\mathbf{A})}$, there exists an infinite series of time $(T_1, T_2, \cdots)$ such that, for any $i$, the probability in $S_2$ at round $T_i$ is at least $1 - \lambda$.*

ACKNOWLEDGMENTS

We thank several anonymous reviewers for their suggestions. This research/project is supported in part by the National Research Foundation, Singapore under its AI Singapore Program (AISG Award No: AISG2-RP-2020-016), NRF 2018 Fellowship NRF-NRFF2018-07, NRF2019-NRF-ANR095 ALIAS grant, grant PIE-SGP-AI-2020-01, AME Programmatic Fund (Grant No. A20H6b0151) from the Agency for Science, Technology and Research (A*STAR) and Provost's Chair Professorship grant RGEPPV2101. Yixin Tao acknowledges ERC Starting Grant ScaleOpt-757481.

REPRODUCIBILITY STATEMENT

We present complete proofs for all theoretical results in this work.

ETHICS STATEMENT

We do not see any ethical or future societal consequences of this work.

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

## A    MOTIVATING EXAMPLE

To explain our mathematical model intuitively, we present the following simple one-dimensional example. Let $f(x) = 2x$, and suppose that $X^0$ is a random variable uniform on the interval $[0, 1]$. Then $X^1 = f(X^0)$ becomes another random variable, which is uniform on $[0, 2]$. Analogously, $X^2 = f(X^1)$ is a new random variable being uniform on $[0, 4]$. Generally speaking, the problem is to understand how the initial random variable $X^0$ evolves under a given deterministic iterated update rule $f$, and to study how the uncertainty evolves using the measure of differential entropy. In our paper, we focus on such systems which arise in the context of learning-in-games.

In a learning-in-game system, $X^0 = (\mathbf{p}^0, \mathbf{q}^0)$ represents a distribution over initial beliefs (initial cumulative payoffs) of the two players, and $X^t = (\mathbf{p}^t, \mathbf{q}^t)$ represents the distribution of cumulative payoffs at time $t$. Such a probabilistic formulation is natural from the perspective of an external analyst (who is not a player of the game), who cannot be certain about what the initial beliefs the players have. For her to proceed with an analysis of the system, a natural approach is to infer the initial distributions from either the past data, or her knowledge about how random external signals influence the initializations. The iterated update rule $f$ in this case is determined by the learning algorithms used by the players, and by the underlying game. For instance, if both players are using MWU in a bimatrix game $(\mathbf{A}, \mathbf{B})$, the update rule $f$ is given by (4).

Unlike the motivating example, for learning-in-game systems the dimensions can be arbitrarily high, and the update rule is non-linear. Thus, deriving the exact evolution of the distribution of $X^t$ is very hard in general. But we can still use the popular statistical measure, differential entropy (DE), to understand the evolution of uncertainty. We show that the DE increases linearly with time for various combinations of learning algorithms and games, including MWU in zero-sum games, OMWU in coordination games, and MWU in certain one-population games. Section 2 has precise descriptions of the learning algorithms and the games. These results can be extended to a broad family of non-zero-sum and non-coordination games, via the Jacobian analysis by Cheung & Tao (2020). The results also extend to gradient ascent, as it is known to be a special case of FTRL.

## B    INTEGRATION BY SUBSTITUTION

A general theorem of integration by substitution is stated below.

**Theorem B.1 (Fremlin (2000), Theorem 263D)** *Let $D \subset \mathbb{R}^d$ be any measurable set, and $\phi : D \to \mathbb{R}^d$ be an injective function differentiable relative to its domain at each point of $D$. For each $x \in D$, let $\mathbf{J}_\phi(x)$ be the Jacobian of $\phi$ relative to $D$ at $x$. Then for any real-valued function $h$ defined on $\phi(D)$,*

$$\int_{\phi(D)} h(x)\, \mathrm{d}x \;=\; \int_D h(\phi(x)) \cdot |\det \mathbf{J}_\phi(x)|\, \mathrm{d}x$$

*if either integral is well-defined, provided that $h(\phi(x)) \cdot |\det \mathbf{J}_\phi(x)|$ is interpreted as zero when $\det \mathbf{J}_\phi(x) = 0$ and $h(\phi(x))$ is undefined.*

In Section 3, we use integration by substitution twice. First, in the above theorem, by setting $h = g$, $\phi = f^{-1}$ and $D = f(\mathcal{X}')$ for any measurable set $\mathcal{X}' \subset X$, we have

$$\int_{\mathcal{X}'} g(x)\, \mathrm{d}x \;=\; \int_{f(\mathcal{X}')} g(f^{-1}(x)) \cdot |\det \mathbf{J}_{f^{-1}}(x)|\, \mathrm{d}x\;.$$

Recall that $g$ is a probability density function of the absolutely continuous random variable $X$, so the LHS of the above equality is same as $\mathbb{P}\left[X \in \mathcal{X}'\right]$. Thus, the integrand of the integral in the RHS is the probability density function of $f(X)$, i.e. $\hat{g}(x) = g(f^{-1}(x)) \cdot |\det \mathbf{J}_{f^{-1}}(x)|$.

To derive (6), in the above theorem, we set $h(x) = -\hat{g}(x) \log \hat{g}(x)$, $\phi = f$, $D = \mathcal{X}$ and hence $\phi(D) = \mathcal{Y}$.

## C    MORE APPLICATIONS

### C.1    OMWU IN TWO-PLAYER COORDINATION GAMES

A two-player game $(\mathbf{A}, \mathbf{B})$ is a *coordination game* if $\mathbf{A} = \mathbf{B}$. Similar to the MWU dynamics in two-player zero-sum games, here we also look at two different areas: $S_1 = \{(p, q)|x(p) \geq \delta$ and $y(q) \geq$

$\delta\}$ and $S_2$, which is the complement to $S_1$, $S_2 = \overline{S_1}$. For a coordination game $(\mathbf{A}, \mathbf{A})$, Cheung & Piliouras (2020) showed that when $(p, q) \in S_1$, then $\det \mathbf{J}_f(x) \geq 1 + \frac{\delta^2 c(A)^2}{8} \cdot \epsilon^2$ when $\epsilon$ is small enough and, when $(p, q) \in S_2$, then $\det \mathbf{J}_f(x) \geq 1 - O(\epsilon^3)$. Therefore, by a calculation which is similar to MWU in two-player zero-sum game, we obtain the following theorems.

**Theorem C.1** *If $\int_{S_1} g^t(\mathbf{p}, \mathbf{q}) \, \mathsf{d}(\mathbf{p}, \mathbf{q}) \geq \lambda$ then the entropy increases by at least $\frac{\delta^2 c(\mathbf{A})^2 \cdot \epsilon^2 / 8}{1 + \delta^2 c(\mathbf{A})^2 \cdot \epsilon^2 / 8} \cdot \lambda - O(\epsilon^3) \cdot (1 - \lambda)$.*

**Theorem C.2** *For any non-trivial two-player coordination games $(\mathbf{A}, \mathbf{A})$, with OMWU dynamics of a small enough step-size $\epsilon$, for any initial distribution with bounded support, and for any $\lambda$ such that $\epsilon = o(\lambda)$, there exists an infinite series of time $(T_1, T_2, \cdots)$, such that the probability in $S_2$ at round $T_i$ is at least $1 - \lambda$.*

### C.2 OMWU IN ONE POPULATION GAMES

As pointed out in Cheung & Piliouras (2020), the continuous analogue of OMWU, in the context of online learning, is as follows. There are $n$ actions. Let $\mathbf{p}(t) \in \mathbb{R}^n$ denote the cumulative payoff vector up to time $t$, and $\mathbf{u}(t) \in \mathbb{R}^n$ denote the instantaneous payoff vector to the $n$ actions at time $t$. The continuous analogue is

$$\dot{\mathbf{p}} = \mathbf{u} + \epsilon \cdot \dot{\mathbf{u}}. \tag{9}$$

In one-population game, $\mathbf{u}(t) = \mathbf{A} \cdot \mathbf{x}(t)$, where for the logit conversion used in MWU and OMWU, $x_k = \exp(p_k) / \sum_\ell \exp(p_\ell)$. It is easy to compute that

$$\frac{\partial x_k}{\partial p_k} = x_k - (x_k)^2 \qquad \text{and} \qquad \frac{\partial x_k}{\partial x_\ell} = -x_k x_\ell \text{ when } \ell \neq k.$$

Then for any $\ell = 1, 2, \ldots, n$, we have

$$\frac{\partial [\mathbf{A}\mathbf{x}]_j}{\partial p_\ell} = \sum_k A_{jk} \cdot \frac{\partial x_k}{\partial p_\ell} = A_{j\ell} x_\ell - \sum_k A_{jk} x_k x_\ell = x_\ell \left( A_{j\ell} - [\mathbf{A}\mathbf{x}]_j \right).$$

Following the notation in MWU case, $\frac{\partial [\mathbf{A}\mathbf{x}]_j}{\partial p_\ell} = R_{j\ell}$. By the chain rule, we have

$$\dot{u}_j = \frac{\mathsf{d}[\mathbf{A}\mathbf{x}]_j}{\mathsf{d}t} = \sum_{\ell=1}^n R_{j\ell} \cdot \dot{p}_\ell.$$

Putting the above equality into (9) yields

$$\dot{p}_j = [\mathbf{A}\mathbf{x}]_j + \epsilon \cdot \sum_{\ell=1}^n R_{j\ell} \cdot \dot{p}_\ell,$$

which is a recurrence of the variables $\dot{p}_1, \dot{p}_2, \ldots, \dot{p}_n$. Unwinding this recurrence yields

$$\dot{p}_j = [\mathbf{A}\mathbf{x}]_j + \epsilon \cdot \sum_{\ell=1}^n R_{j\ell} \cdot [\mathbf{A}\mathbf{x}]_\ell + \mathcal{O}(\epsilon^2).$$

The above system, upon Euler discretization to a discrete-time dynamical system with time interval $\epsilon$, becomes

$$\forall j, \quad p_j^{t+1} = p_j^t + \epsilon \cdot [\mathbf{A}\mathbf{x}]_j + \epsilon^2 \cdot \sum_{\ell=1}^n R_{j\ell} \cdot [\mathbf{A}\mathbf{x}]_\ell + \mathcal{O}(\epsilon^3). \tag{10}$$

Observe that on the RHS of (10), the first two terms coincide with that in the MWU case. The remaining terms can be viewed as a *second-order perturbation*. Consequently, the Jacobian of the system equation 10 is a second-order perturbation of the Jacobian for the MWU case, and hence the constant and first-order terms of the determinants of these two Jacobians are the same, which is $\sum_{j=1}^n R_{jj}$.

Therefore, for the case that $A_{kk} + A_{k'k'} \geq A_{kk'} + A_{k'k}$ for all $k$ and $k'$ and there exist $k$ and $k'$ such that the inequality is strict, we obtain the following theorems. Recall that $S_1 = \{\mathbf{p} | \mathbf{x}(\mathbf{p}) \geq \delta\}$ and $S_2$, which is the complement to $S_1$, $S_2 = \overline{S_1}$; and $s(\mathbf{A}) = \max_{k,k'} \{A_{kk} + A_{k'k'} - A_{kk'} - A_{k'k}\}$.

**Theorem C.3** *If $\int_{S_1} g^t(\mathbf{p}) \, d\mathbf{p} \geq \lambda$, then the entropy increases by at least $\frac{\delta^2 s(\mathbf{A})\epsilon/2}{1+\delta^2 s(\mathbf{A})\epsilon/2} \cdot \lambda - O(\epsilon^2) \cdot (1-\lambda)$ for a small enough step size $\epsilon$.*

**Theorem C.4** *In an one-population game $\mathbf{A}$ such that $A_{kk} + A_{k'k'} \geq A_{kk'} + A_{k'k}$ for all $k$ and $k'$ and there exist $k$ and $k'$ such that the inequality is strict, with OMWU dynamics with a small enough step-size $\epsilon$, for any initial distribution with bounded support, and for any $\lambda$ such that $\epsilon = o(\lambda)$, there exists an infinite series of time $(T_1, T_2, \cdots)$ such that, for any $i$, the probability in $S_2$ at round $T_i$ is no less than $1 - \lambda$.*

For the case that $A_{kk} + A_{k'k'} = A_{kk'} + A_{k'k}$ for all $k$ and $k'$, we consider the second-order term. The second-order terms of the two determinants (MWU and OMWU) are not the same though. The second-order term of the determinant of Jacobian for the OMWU case is

$$\sum_{1 \leq j < k \leq n} R_{jj} R_{kk} \; - \; \sum_{1 \leq j < k \leq n} R_{jk} R_{kj} \; + \; \sum_{j=1}^{n} \sum_{\ell=1}^{n} R_{j\ell} R_{\ell j} \; + \; \sum_{j=1}^{n} \sum_{\ell=1}^{n} \frac{\partial R_{j\ell}}{\partial p_j} \cdot [\mathbf{Ax}]_\ell.$$

Observe that the first two summations are just the same as those in the MWU case. Next, a straight-forward calculation gives

$$\frac{\partial R_{j\ell}}{\partial p_j} \; = \; R_{jj} \cdot \mathbf{1}_{j=\ell} - x_j R_{j\ell} - x_\ell R_{jj},$$

where $\mathbf{1}_{j=\ell} = 1$ if $j = \ell$, and 0 otherwise. Hence

$$\sum_{j=1}^{n} \sum_{\ell=1}^{n} \frac{\partial R_{j\ell}}{\partial p_j} \cdot [\mathbf{Ax}]_\ell \; = \; \sum_{j=1}^{n} R_{jj} \cdot [\mathbf{Ax}]_j \; - \; \sum_{j=1}^{n} \sum_{\ell=1}^{n} (x_j R_{j\ell} + x_\ell R_{jj}) \cdot [\mathbf{Ax}]_j$$

$$= \; \sum_{j=1}^{n} R_{jj} \cdot [\mathbf{Ax}]_j \; - \; \sum_{j=1}^{n} x_j [\mathbf{Ax}]_j \sum_{\ell=1}^{n} R_{j\ell} \; - \; \sum_{j=1}^{n} R_{jj} [\mathbf{Ax}]_j \sum_{\ell=1}^{n} x_\ell.$$

Since $\sum_{\ell=1}^{n} x_\ell = 1$ and $\sum_{\ell=1}^{n} R_{j\ell} = 0$, we have $\sum_{j=1}^{n} \sum_{\ell=1}^{n} \frac{\partial R_{j\ell}}{\partial p_j} \cdot [\mathbf{Ax}]_\ell = 0$. To conclude, the second-order term is

$$\sum_{1 \leq j < k \leq n} R_{jj} R_{kk} \; - \; \sum_{1 \leq j < k \leq n} R_{jk} R_{kj} \; + \; \sum_{j=1}^{n} \sum_{\ell=1}^{n} R_{j\ell} R_{\ell j}$$

$$= \; \sum_{j=1}^{n} (R_{jj})^2 \; + \; \sum_{1 \leq j < k \leq n} R_{jj} R_{kk} \; + \; \sum_{1 \leq j < k \leq n} R_{jk} R_{kj}$$

$$= \; \left( \sum_{j=1}^{n} R_{jj} \right)^2 \; - \; \sum_{1 \leq j < k \leq n} R_{jj} R_{kk} \; + \; \sum_{1 \leq j < k \leq n} R_{jk} R_{kj}$$

Note that when $A_{kk} + A_{k'k'} = A_{kk'} + A_{k'k}$ for all $k$ and $k'$, $\sum_j R_{jj} = 0$. Therefore, the second order term of OMWU is the negative of the second order term of MWU, which will be smaller than $-\frac{\delta^2 c(A)}{8}$.

**Theorem C.5** *In an one-population game $\mathbf{A}$ such that $A_{kk} + A_{k'k'} \geq A_{kk'} + A_{k'k}$ for all $k$ and $k'$, with OMWU dynamics, if $\int_{p \in S_1} g(p) \, dp \geq \lambda$, then the entropy decreases by at least $\frac{\delta^2 c(\mathbf{A})^2 \epsilon^2}{1 + \delta^2 c(\mathbf{A})^2 \epsilon^2} \cdot \lambda - O(\epsilon^3) \cdot (1-\lambda)$ for a small enough step size $\epsilon$.*

## C.3 PERTURBATIONS TO PAYOFFS

In the main body, we focus on two-player zero-sum games. In this appendix, we discuss the Jacobian analysis for two-player general games based on Cheung & Tao (2020), and explain why it is robust against perturbations.

First, we define the following function on the cumulative payoff space of two-player games, which depends on a matrix $\mathbf{M} \in \mathbb{R}^{n \times m}$.[5]

$$L_{\mathbf{M}}(\mathbf{p}, \mathbf{q}) := \frac{1}{4} \sum_{\substack{1 \le j, j' \le n \\ 1 \le k, k' \le m}} x_j(\mathbf{p}) x_{j'}(\mathbf{p}) y_k(\mathbf{q}) y_{k'}(\mathbf{q}) \cdot (M_{jk} + M_{j'k'} - M_{jk'} - M_{j'k})^2,$$

where $x_j, x_{j'}, y_k, y_{k'}$ is as defined in (2).

For any general game $(\mathbf{A}, \mathbf{B})$, we can decompose it into the sum of a zero-sum game and a coordination game as $(\mathbf{A}, \mathbf{B}) = (\mathbf{Z}, -\mathbf{Z}) + (\mathbf{C}, \mathbf{C})$, where $\mathbf{Z} = (\mathbf{A} - \mathbf{B})/2$ and $\mathbf{C} = (\mathbf{A} + \mathbf{B})/2$. Cheung & Tao (2020) showed that the determinant of the Jacobian matrix of MWU with step-size $\epsilon$ in the game $(\mathbf{A}, \mathbf{B})$ is

$$1 + (L_{\mathbf{Z}}(\mathbf{p}, \mathbf{q}) - L_{\mathbf{C}}(\mathbf{p}, \mathbf{q})) \epsilon^2 + \mathcal{O}(\epsilon^4).$$

We rewrite $L_{\mathbf{Z}}(\mathbf{p}, \mathbf{q}) - L_{\mathbf{C}}(\mathbf{p}, \mathbf{q})$ as

$$\frac{1}{4} \sum_{\substack{1 \le j, j' \le n \\ 1 \le k, k' \le m}} x_j(\mathbf{p}) x_{j'}(\mathbf{p}) y_k(\mathbf{q}) y_{k'}(\mathbf{q}) \cdot (A_{jk} + A_{j'k'} - A_{jk'} - A_{j'k})(B_{jk'} + B_{j'k} - B_{jk} - B_{j'k'}).$$

Thus, if

$$\max_{j, j', k, k'} [(A_{jk} + A_{j'k'} - A_{jk'} - A_{j'k})(B_{jk'} + B_{j'k} - B_{jk} - B_{j'k'})] \ge \overline{L} > 0 \qquad (11)$$

and

$$\min_{j, j', k, k'} [(A_{jk} + A_{j'k'} - A_{jk'} - A_{j'k})(B_{jk'} + B_{j'k} - B_{jk} - B_{j'k'})] \ge 0, \qquad (12)$$

then in the domain $S_1 = \{(\mathbf{p}, \mathbf{q}) \mid \forall j, k, \ x_j(\mathbf{p}) \ge \delta \text{ and } y_k(\mathbf{q}) \ge \delta\}$, the determinant is at least $1 + \delta^4 \overline{L} \epsilon^2 / 8$ for all sufficiently small $\epsilon$.

Since the product $(A_{jk} + A_{j'k'} - A_{jk'} - A_{j'k})(B_{jk} + B_{j'k'} - B_{jk'} - B_{j'k})$ depends on $\mathbf{A}, \mathbf{B}$ continuously, it is natural to expect that when $\mathbf{A}, \mathbf{B}$ are slightly perturbed, the value of the product changes also slightly. To make this precise, we suppose that in each round $t$, the actual payoff matrices are $(\mathbf{A}^t, \mathbf{B}^t)$, and their entries are perturbations from $(\mathbf{A}, \mathbf{B})$ by at most an absolute value of $\beta \le 2$.[6] Recall that the entries in $\mathbf{A}, \mathbf{B}$ are bounded in the interval $[-1, 1]$, so the absolute values of $A_{jk} + A_{j'k'} - A_{jk'} - A_{j'k}$ and $B_{jk'} + B_{j'k} - B_{jk} - B_{j'k'}$ are bounded by 4, while the perturbations to their values are bounded by $4\beta$. By the lemma below, for the game $(\mathbf{A}^t, \mathbf{B}^t)$, the bound in (11) still hold by replacing its lower bound with $\overline{L} - 34\beta$, and the bound in (12) still hold by replacing its lower bound with $-34\beta$. Thus the determinant of the Jacobian matrix in $S_1$ is at least $1 + (\delta^4 \overline{L}/4 - 34\beta) \epsilon^2 / 2 \ge 1 + \delta^4 \overline{L} \epsilon^2 / 16$ if $\beta \le \delta^4 \overline{L}/272$.

**Lemma C.6** *If $\beta \le 2$, $|a|, |b| \le 4$ and $|a' - a|, |b' - b| \le 4\beta$, then $a'b' \ge ab - 34\beta$.*

**Proof:** Let $a' - a = \beta_a$, $b' - b = \beta_b$. Then $a'b' - ab = b\beta_a + a\beta_b + \beta_a \beta_b \ge -16\beta - 16\beta - \beta^2 \ge -34\beta$. $\qquad \square$

# D    PLOTS OF UNCERTAINTY EVOLUTION OF MWU IN ONE-POPULATION GAME

---

[5]This function was denoted by $C_{(\mathbf{M}, -\mathbf{M})}(\mathbf{p}, \mathbf{q})$ in Cheung & Tao (2020), but to avoid confusion and cluster of notation, we change it to $L_{\mathbf{M}}(\mathbf{p}, \mathbf{q})$.

[6]We assume that the entries in both $(\mathbf{A}, \mathbf{B})$ and $(\mathbf{A}^t, \mathbf{B}^t)$ are in the interval $[-1, 1]$, so $\beta$ is at most 2.

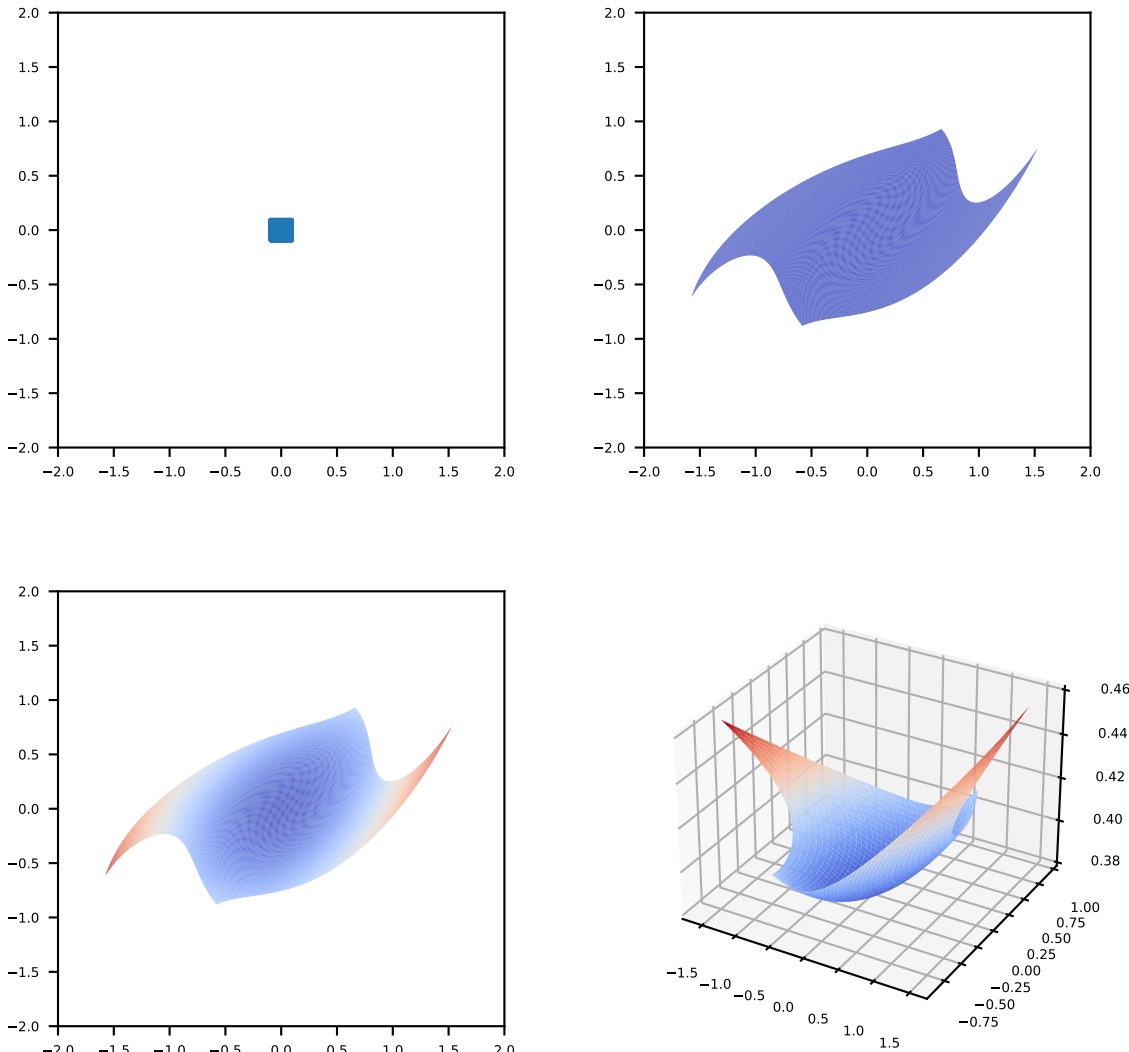

Figure 2: We plot the evolution of uncertainty for MWU in an one-population game specified by the matrix $\mathbf{A} = \begin{bmatrix} 0.1 & 1.0 & -1.0 \\ -1.0 & 0.1 & 1.0 \\ 1.0 & -1.0 & 0.1 \end{bmatrix}$, with step-size $\epsilon = 0.01$. As we did in Figure 2, in the top left plot is the initial distribution, which is uniform in the small square. The top right plot shows the "possibility" plot after 6000 steps, and the bottom left plot shows the corresponding heat-map plot of probability densities.

