# OpenReview forum: "The Evolution of Uncertainty of Learning in Games"
_ICLR.cc/2022/Conference — ICLR 2022 Poster_

### Official Review · Reviewer_2Gu6 · 2021-10-31

**Correctness:** 4
**Technical Novelty And Significance:** 3
**Empirical Novelty And Significance:** Not applicable
**Recommendation:** 6
**Confidence:** 3

**Main Review:**

Learning in games is a relevant and active area of research, and I believe the contributions of this paper are original and novel with respect to previous works. Moreover, the paper is well-written and the authors did a good job in illustrating the motivations and putting their work into context. As mentioned by the authors, the obtained results are orthogonal w.r.t. past works that study non-convergence in games, as the proposed approach quantifies the increase of unpredictability via the notion of differential entropy.

In my opinion, however, the paper has also some weaknesses:
The paper lacks experimental results which perhaps could better illustrate the studied phenomena (e.g., the obtained DE rates).  Moreover, the obtained theoretical results, albeit novel, revolve around a few technical arguments and sound a bit repetitive.

**Summary Of The Paper:**

The paper studies the evolution of uncertainty in multi-agent game dynamics. More specifically, it studies how the probability distribution over the players' cumulative payoffs evolves as players use typical online learning algorithms to play the game.
The game uncertainty is quantified by the notion of Differential Entropy (DE) of such distribution, a quantity related to the Jacobian of the game dynamics. Authors show that DE increases linearly with time for a set of games including two-player zero-sum, coordination games, and population games, confirming the negative convergence results obtained in past works.

**Summary Of The Review:**

To summarize, although the paper could improve from additional experiments and the technical contributions are limited, I have appreciated the motivations and model proposed by the authors, as well as the obtained theoretical results.
Hence, my accept score.

---

> ### Author Response · Authors · 2021-11-23
> **Response to Reviewer 2Gu6**
>
> We thank the reviewer for their work and appreciate their support for our paper. We add one more experiment for MWU in an one-population game (page 15). I think that it is important to note that such findings provide new, concrete insights in the rather well developed field of evolutionary game theory (see very well cited books such as [1-4]). The perspective of evolution under uncertainty clearly is a very natural one in these settings where the "mixed-strategy" encodes the state of a population (e.g. what percentages of animals in a population has genetic trait A, B, or C or what percentage of a large populations of agents uses strategy A, B or C). Such information can only be known with limited certainty and how this uncertainty evolves over time is a critical but so far unexplored feature of these models.
>
> > Moreover, the obtained theoretical results, albeit novel, revolve around a few technical arguments and sound a bit repetitive.
>
> We agree that the technical arguments that lead to the novel theoretical results in Sections 4.1 and 4.2 are similar. To us, this is more of a strength than a weakness, since it demonstrates that our technical arguments can be applicable to more than one settings, and are potentially useful in other settings (such as possibly other learning dynamics, different classes of games, smooth/differentiable games, markets, etc). The robustness of the applicability of our techniques is a promising feature in this direction.
>
> 1. Sandholm WH Population games and evolutionary dynamics 2010
> 2. Hofbauer J, Sigmund K. Evolutionary games and population dynamics 1998.
> 3. Weibull J.W. Evolutionary game theory 1997
> 4. Nowak M.A. Evolutionary Dynamics: Exploring the Equations of Life 2007

---

### Official Review · Reviewer_Xwzj · 2021-11-01

**Correctness:** 4
**Technical Novelty And Significance:** 4
**Empirical Novelty And Significance:** Not applicable
**Recommendation:** 6
**Confidence:** 2

**Main Review:**

The paper studies the dynamics of online learning. Different from the standard perspective, the paper investigates the behavior of a probability distribution over a set of initial conditions.
The paper provides a novel theoretical insight on how this probability distribution over initial conditions envolve. The technical/theoretical aspects of this paper seem to be rigorous.

1. Second line below eq (1), what is q^t here? should be p^t?
2. In eq (1), the agent updates the cumulative payoff vector using the exact payoff at the current time step. I’m curious whether the main results still hold if the agent uses the empirical payoffs till to current time step?
3. The authors characterize the evolution of the differential entropy of the distribution of cumulative payoff vectors. When the probability distribution over the initial conditions is discrete, which is often seen in practice, can the authors still achieve the same characterization for Shannon entropy?


**Summary Of The Paper:**

The paper studies how the uncertainty of the initial cumulative payoff vectors evolves in the process of learning in games. Looking to the differential entropy, the authors show that for a broad range of learning-in-game systems, including two-player matrix games and one-population games, the differential entropy of the distribution over the cumulative payoff vectors increases linearly with time in two learning algorithms (Multiplicative Weights Update, and Optimistic Multiplicative Weights Update).

**Summary Of The Review:**

I’m not super familiar with the related literature of this work. But as far as I’m aware, the perspective adopted in this paper on studying the dynamics of online learning in games is original to me. The characterization on the evolution of cumulative payoff vectors seems pretty novel to me. I feel the results obtained in this work may have implications on other studies on the behavior dynamics of online learning in games, especially for those who need random initial conditions.

---

> ### Author Response · Authors · 2021-11-23
> **Response to Reviewer Xwzj**
>
> We thank the reviewer for their work and appreciate their support for our paper. We hope the following response can fully address your comments. Please let us know if you have any further comments.
>
> > Second line below eq (1), what is q^t here? should be p^t?
>
> Yes, you're right. It is fixed now.
>
> > In eq (1), the agent updates the cumulative payoff vector using the exact payoff at the current time step. I’m curious whether the main results still hold if the agent uses the empirical payoffs till to current time step?
>
> This is a very interesting question indeed. A recently posted arxiv paper [1] argues that in the case where you start MWU with a single point measure then its the stochastic evolution results in measures that concentrate on the corners of the state-space. Although this does not offer a proper quantitative statement about the evolution of DE it definitely agrees in spirit with our findings. We aim to investigate this setting in the future and offer a more detailed understanding of it. Early experiments that we have performed in this setting indicate that the differential entropy also increases over time.
>
> > The authors characterize the evolution of the differential entropy of the distribution of cumulative payoff vectors. When the probability distribution over the initial conditions is discrete, which is often seen in practice, can the authors still achieve the same characterization for Shannon entropy?
>
> The Shannon entropy does not increase if the probability distribution over the initial conditions is discrete. This is due to a well-known property of Shannon entropy that it is preserved under bijective mapping. In our case, when the step size is small (less than 1/4), the MWU dynamics is a bijective mapping from time t to time t+1.
>
> [1] Bailey et. al. Stochastic Multiplicative Weights Updates in Zero-Sum Games arXiv:2110.02134

---

### Official Review · Reviewer_8PmF · 2021-11-01

**Correctness:** 4
**Technical Novelty And Significance:** 2
**Empirical Novelty And Significance:** Not applicable
**Recommendation:** 8
**Confidence:** 3

**Main Review:**

Strengths
 - Enables looking at differential entropy, and not just the volume, of a distribution of strategies for a game.
 - Strengthens the approach of previous work by splitting the space into a boundary region and interior region and analyzing the change in probability mass between the two.
 - Contributes a possibly novel analysis of the determinant of a Jacobian of a population game
 - Results are claimed to be robust to small perturbations in the game.

Weaknesses
 - Is differential entropy (DE) the most appropriate way to index the deviation of the strategy vector from equilibrium?  This paper could provide arguments for why DE makes more sense than other statistics that could be considered, e.g. determinant of covariance matrix, trace of covariance, Nash gap (max difference between a player's payoff and the payoff for their best response holding other players fixed.)
 - Is the use of DE key to the results on non zero-sum games, or could similar results have been obtained through volume analysis?   This paper could make it clear which novel results are exclusive to the choice of using DE.
 - The significance of the results, in spite of a brief reference to the "grand escape" (which is not further elaborated outside of the introduction), is not made clear to the reader.  This paper could describe in greater detail what happens to make the entropy increase slow down.

Minor comments
 - Does there exist a reference for the derivation for eq. 6?  It seems to be known in the literature, e.g. see Duvenald et al. 2016 "Early Stopping as Nonparametric Variational Inference"
 - pg. 2 "more probably to occur" -> "more probable to occur"
 - pg. 4 after eq (1) "depending on the value of qt" -> "depending on the value of pt"

**Summary Of The Paper:**

This paper extends the existing line of research of the dynamics of multiplicative weights update and similar algorithms for games.  It shows that for zero-sum two-player games and for population games satisfying certain conditions, that the entropy increases linearly as long as the strategies are far from the distribution; this implies that the strategies concentrate near the boundary in the long-run, in a sense that the paper formalizes.  Compared to previous work, this analysis applies to games that are not zero-sum--specifically, population games that satisfy a certain condition on the payoffs.

**Summary Of The Review:**

This paper adds some new results extending previous work that established divergence from Nash equilibrium for a class of learning algorithms for games.  I was not convinced of the main argument of the paper, which claims that the advantage of using differential entropy to analyze the learning dynamics.  What seemed more novel was the analysis of the Jacobian of the population game.  Hence, overall I am recommending weak acceptance.  If I were to be convinced that the use of differential entropy is a promising approach for the field, then I would raise my score.

---

> ### Author Response · Authors · 2021-11-23
> **Response to Reviewer 8PmF**
>
> We thank the reviewer for their work and appreciate their support for our paper. We address your comments as below.
>
> >Is differential entropy (DE) the most appropriate way to index the deviation of the strategy vector from equilibrium? This paper could provide arguments for why DE makes more sense than other statistics that could be considered, e.g. determinant of covariance matrix, trace of covariance, Nash gap (max difference between a player's payoff and the payoff for their best response holding other players fixed.)
>
> It is true that there are several statistical measures which can capture uncertainty evolution, e.g. variance (or determinant of covariance matrix in high dimension). We choose to use DE for two key reasons. First, the analysis of the evolution of DE is easier; to analyze variance, we need to keep track of the changes of both the mean of the random variables and the probability densities, which seem more involved technically. But to analyze DE, we only need to focus on the Jacobians. Second, DE is actually a lower bound of the determinant of covariance matrix and the trace of the covariance (Cover and Thomas, pp. 254): for any distribution in $\mathbb{R}^n$, its DE is at most
> $$\frac 12 log (2 \pi e)^n + \frac 12 \log \text{(determinant of covariance matrix)} \le \frac 12 log (2 \pi e)^n + \frac n2 \log (\text{(trace of covariance matrix)}/n).$$
> (The final inequality is due to the AM-GM inequality, and noting that determinant is product of eigenvalues while trace is sum of eigenvalues.) Therefore, a linear increase of DE implies a linear increasing lower bound on log (determinant of covariance matrix) and log (trace of covariance matrix).
>
> It is less clear how to analyze the Nash gap in our setting, but it is an interesting direction to explore.
>
> Another important reason that DE is an interesting tool, is due to its connections to the relative information entropy, also known as the Kullback–Leibler divergence. Understanding the properties of KL-divergence between the NE of a zero-sum/coordination game and the state has been a powerful tool in the case of understanding the evolution of single point orbits e.g.,[1, 2, 3, 4, 5] for MWU/OMWU and variants (discrete/continuous time). We hope that such ideas could inspire similar progress in our understanding of evolution of more general measures (learning with uncertainty) instead of merely the evolution of Dirac measures (i.e. points).
>
> 1. Piliouras G. and Shamma J. Optimization Despite Chaos: Convex Relaxations to Complex Limit Sets via Poincaré Recurrence. ACM-SIAM Symposium on Discrete Algorithms (SODA), 2018.
> 2. Mertikopoulos P. et. al . Cycles in Adversarial Regularized Learning. ACM-SIAM Symposium on Discrete Algorithms (SODA), 2018.
> 3. Bailey J. et al. Multiplicative Weight Update in Zero-Sum Games. ACM Conference on Economics and Computation (EC), 2018.
> 4. Daskalakis et al. Last-iterate convergence: Zero-sum games and constrained min-max optimization. ITCS 19.
> 5. Nagarajan S.G. et al. From Chaos to Order: Symmetry and Conservation Laws in Game Dynamics. ICML, 2020.
>
> >Is the use of DE key to the results on non zero-sum games, or could similar results have been obtained through volume analysis? This paper could make it clear which novel results are exclusive to the choice of using DE.
>
> To us, volume analysis and DE analysis are the natural approaches to analyzing two different models, although they do share some similarities arithematically. Compared to volume analysis, as what we said in the paper, "Our model and analysis can be viewed as an a strengthening of volume analysis. Volume analysis focuses on whether it is possible to reach a state or not, whereas in our model we are also concerned with how probable/likely it is to reach a state."
>
> More specifically, for two-person zero-sum games, in Theorem 4.5, by focusing on differential entropy, we can have a lower bound on the probability that the initial set escapes to the boundary, while by volume analysis we can only conclude that there is always a point in the initial set that escapes to the boundary.
>
> >The significance of the results, in spite of a brief reference to the "grand escape" (which is not further elaborated outside of the introduction), is not made clear to the reader. This paper could describe in greater detail what happens to make the entropy increase slow down.
>
> We apologize for not discussing the "grand escape" in greater detail. By "grand escape", we are referring to Theorem 4.5.
>
> >Does there exist a reference for the derivation for eq. 6? It seems to be known in the literature, e.g. see Duvenald et al. 2016 "Early Stopping as Nonparametric Variational Inference"
>
> Thanks for pointing us to this literature. We added the citation at the top of page 6.
>
> >pg. 2 "more probably to occur" -> "more probable to occur"
> >pg. 4 after eq (1) "depending on the value of qt" -> "depending on the value of pt"
>
> Both fixed. Thank you!

---

> > ### Comment · Reviewer_8PmF · 2021-11-25
> > **Thanks for your response**
> >
> > Thank you for your response.  I'm raising my score to 7. (but since 7 is not available, I will rate it as 8)

---

### Official Review · Reviewer_8gHb · 2021-11-03

**Correctness:** 3
**Technical Novelty And Significance:** 2
**Empirical Novelty And Significance:** Not applicable
**Recommendation:** 5
**Confidence:** 2

**Main Review:**

Questions:
* Fig. 1 is extremely unclear, and it would be better to explain the meaning of each sub-fig in the caption or main text.
* I think this paper is not easy to follow, and many clarifications are needed. For example, a general problem setting before the model section would be useful. Why choose MWU in zero-sum games, OMWU in coordination games? Would the same conclusions still hold when using other learning algorithms in games like gradient ascent, joint action learner?

**Summary Of The Paper:**

This paper proposes a differential entropy framework to quantify the evolutionary stability of learning in games with different initial conditions. The paper finds that differential entropy of these learning-in-game systems increases linearly with time for Multiplicative Weights Update (MWU) or Follow- the-Regularized-Leader (FTRL) algorithms in zero-sum games.


**Summary Of The Review:**

Investigating the uncertainty/stability during the learning in games with different initial conditions is an interesting problem. But from my understanding, this problem has not been well formalized in the paper, and many additional clarifications are required to make the people unfamiliar with this research problem get the paper's key points.

---

> ### Author Response · Authors · 2021-11-23
> **Response to Reviewer 8gHb**
>
> Thank you for taking the time to review this paper. We address your comments as below.
>
> First, we'd like to explain our model by using the following simple 1-dimensional motivating example. Let $f(x) = 2x$, and suppose that $X^0$ is a random variable uniform on the interval $[0,1]$. Then $X^1 = f(X^0)$ becomes another random variable, which is uniform on $[0,2]$. Analogously, $X^2 = f(X^1)$ is a new random variable being uniform on $[0,4]$. Generally speaking, the problem is to understand how the initial random variable $X^0$ evolves under a given deterministic iterated update rule $f$, and to study how the uncertainty evolves using the measure of differential entropy. In our paper, we focus on such systems which arise in the context of learning-in-games.
>
> In a learning-in-game system, $X^0 = (p^0,q^0)$ represents a distribution over initial beliefs (initial cumulative payoffs) of the two players, and $X^t = (p^t,q^t)$ represents the distribution of cumulative payoffs at time $t$. Such a probabilistic formulation is natural from the perspective of an external analyst (who is not a player of the game), who cannot be certain about what the initial beliefs the players have. For her to proceed with an analysis of the system, a natural approach is to infer the initial distributions from either the past data, or her knowledge about how random external signals influence the initializations. The iterated update rule f in this case is determined by the learning algorithms used by the players, and by the underlying game. For instance, if both players are using MWU in a bimatrix game $(\mathbf{A},\mathbf{B})$, the update rule f is given by Eqn. (4) in page 5 of our paper.
>
> Unlike the motivating example, for learning-in-game systems the dimensions can be arbitrarily high, and the update rule $f$ is non-linear. Thus, deriving the exact evolution of the distribution of $X^t$ is very hard in general. But we can still use the popular statistical measure, differential entropy (DE), to understand the evolution of uncertainty. We show that the DE increases linearly with time for various combinations of learning algorithms and games, including MWU in zero-sum games, OMWU in coordination games, and MWU in certain one-population games. Section 2 has precise descriptions of the learning algorithms and the games. These results can be extended to a broad family of non-zero-sum and non-coordination games, via the Jacobian analysis by Cheung and Tao (ICLR 2021). The results also extend to gradient ascent, as it is known to be a special case of FTRL. Thanks for pointing out Joint Action Learner is another learning algorithm of interest, we will look into it and see if our analytic technique can be applicable.
>
> Now we are ready to explain what Figure 1 illuminates, which we also discussed in the last paragraph on page 2. Here, we concern a standard instance of a zero-sum game called Matching Pennies, and both players use the MWU learning algorithm. The top left plot is a heatmap of $X^0$, which is uniform over the small square in the middle. The bottom left plot is the heatmap of $X^{40000}$. The red (warm) part is the region with higher probability densities, while the blue (cool) part has lower probability densities. The top right plot is coming from volume analysis, which ignores the varying probability densities across different regions, and only concerns about ``possibility'' --- the probability density of some point may be very low but non-zero, but in the volume analysis perspective, it is possible to occur, so it will be included in the top right plot as with any point that has a much higher probability density.
>
> Thank you again for your questions and suggestions. We have added Appendix A with some of these discussion points as we believe that they will be useful for other readers as well.

---

### Decision · Program_Chairs · 2022-01-20

**Decision:**

Accept (Poster)

**Comment:**

This paper examines the evolution of densities of initial conditions under the multiplicative weights update rule for learning in two-player zero-sum games. Specifically, the authors estimate the differential entropy (DE) of a density of initial conditions as it evolves over time (what they call "uncertainty"), and they show that (a) as long as the density of states assigns sufficient mass to all strategies, its DE will increase; and (b) the density of states will get arbitrarily close to the boundary of the state space infinitely often (i.e., at least one pure strategy will be employed with arbitrarily small probability infinitely often). The authors also apply these results to a population-like model of learning as well as an optimistic variant of the MWU protocol (the latter in the supplement).

The paper was extensively discussed during the review/rebuttal phase. While the reviewers appreciated the conceptual contributions of the paper, they also identified certain technical shortcomings that were only partially addressed by the authors. One of these issues concerned the possibility that the density of initial conditions may exhibit singularities, in which case the DE may fail to be well-defined. As a result, one of the reviewers indicated an intent to downgrade their score from "8" to "3" due to concerns on the correctness of the results presented in the paper.

After discussing with both the authors and the reviewers, my view is that the merits of the paper outweigh its flaws, so I am making an "accept" recommendation. At the same time, there is a number of revisions that the authors will have to undertake in the camera-ready version of their paper:

1. The authors need to be more careful with their assumptions and notation. The reviewers already indicated a number of glitches, most of them easily fixable (so they are not of particular concern). On the other hand, the issue of whether the initial density of states becomes singular or not is more subtle and led one of the reviewers to drastically change their evaluation of the paper.

  The problem here is that the authors are not being precise in their assumptions for $g^1$ and its support, and this confusion remained throughout the discussion: the authors are looking at distributions that are "smooth with bounded support", but this does not exclude singularities. The counterexample given to the authors was a random variable $X$ supported on $\mathcal{X} = (0,1)$ with density $g(x) = 1/(2\sqrt{x})$; this density has bounded support and it is smooth on its support, but it is not itself bounded. [There is an ambiguity here in whether the authors are considering the support to be closed or not.]

  The issue for the initial density can be trivially fixed by asking that $g^1$ be itself bounded (or smooth over the closed support, or any other similar statement). However, even if this is assumed for $g^1$, the density at some later time $t$ could, a priori, become singular (incidentally, this is a problem that arises frequently in the study of densities that evolve over time, e.g., as in optimal transport). Thus, even an explicit assumption for $g^1$ does not suffice to ensure that $g^t$ does not develop singularities in future stages. [Incidentally, the authors' reply that the singularity has measure zero and therefore does not contribute to the integral misses the heart of the matter (and raises concerns about the authors' overall treatment of this question): the function $g(x) = (\log 2) \big/ (x \log^2x)$ has infinite differential entropy over $(0,1/2)$ even though it is a smooth density over $(0,1/2)$.]

  To be clear, I do not believe that blow-ups actuall occur in the authors' model, but there is still something that needs to be shown here. However, since it is impossible to check an argument or proof at this stage (and I do not think it would be fair to let this stand in the way of accepting the paper), the authors should instead revise their paper to add as an **explicit** assumption that $g^t$ has bounded support and is bounded over its support (or clarify whether they take the support to be closed or not).

1. Another concern revolves around the use of the word "uncertainty" to describe the basic premise of the paper. In the authors' model, this does not refer to uncertainty among the learners (all their observations are perfectly certain and deterministic), so it is not used in the sense that is standard in game theory and learning (cf. the classic works of Bertsekas, Dekel, Fudenberg, Tsitsiklis, and many others). Instead, the authors' use of the word seems to refer to some "outside spectator" who can only partially guess the players' initial conditions, and tries to guess the evolution of the players' mixed strategies (but still has full information about the learning model that players use, its parameters, etc.). However, this model is not fleshed out in sufficient detail by the authors, so the term "uncertainty" does not seem appropriate here.

  During the rebuttal phase, the authors argued that the goal of their paper is "bringing the notion of DE to machine learning audience's attention as a measure of uncertainty, explaining how the change of DE is related to the Jacobian of the underlying dynamical systems" and they asked "that [the paper's] title remains as is". While I am sympathetic to the authors' request, the fact remains that the current title (and part of the discussion in the abstract) is not representative of the paper.

  Given the authors' stated objective, the simplest solution would be to frame the paper as the "evolution of differential entropy under..." or the "evolution of spectator/observer uncertainty" or something of the sort. Both titles carry more information and, based on the authors' input, are more appropriate for the range of ideas the authors wish to convey – but simply saying "uncertainty" goes against the established terminology of the field.

Overall, I would urge the authors to avoid vague/ambiguous terminology and statements, and focus instead on exact mathematical definitions that are not open to interpretation. The ideas presented in the paper are interesting and fresh, so they deserve a likewise sharp and precise treatment.